# Protective Role of Melatonin and Its Metabolites in Skin Aging

**DOI:** 10.3390/ijms23031238

**Published:** 2022-01-22

**Authors:** Georgeta Bocheva, Radomir M. Slominski, Zorica Janjetovic, Tae-Kang Kim, Markus Böhm, Kerstin Steinbrink, Russel J. Reiter, Konrad Kleszczyński, Andrzej T. Slominski

**Affiliations:** 1Department of Pharmacology and Toxicology, Medical University of Sofia, 1431 Sofia, Bulgaria; 2Department of Dermatology, University of Alabama at Birmingham, Birmingham, AL 35294, USA; radomir.slominski@gmail.com (R.M.S.); zjanjetovic@uabmc.edu (Z.J.); tkkim4567@gmail.com (T.-K.K.); 3Graduate Biomedical Sciences Program, University of Alabama at Birmingham, Birmingham, AL 35294, USA; 4Department of Dermatology, University of Münster, Von-Esmarch-Str. 58, 48149 Münster, Germany; boehmma@ukmuenster.de (M.B.); kerstin.steinbrink@ukmuenster.de (K.S.); konrad.kleszczynski@ukmuenster.de (K.K.); 5Department of Cellular and Structural Biology, UT Health Science Center, San Antonio, TX 78229, USA; reiter@uthscsa.edu; 6Pathology and Laboratory Medicine Service, VA Medical Center, Birmingham, AL 35294, USA

**Keywords:** melatonin, AFMK, skin aging, photoaging, UV radiation, oxidative stress, anti-aging properties

## Abstract

The skin, being the largest organ in the human body, is exposed to the environment and suffers from both intrinsic and extrinsic aging factors. The skin aging process is characterized by several clinical features such as wrinkling, loss of elasticity, and rough-textured appearance. This complex process is accompanied with phenotypic and functional changes in cutaneous and immune cells, as well as structural and functional disturbances in extracellular matrix components such as collagens and elastin. Because skin health is considered one of the principal factors representing overall “well-being” and the perception of “health” in humans, several anti-aging strategies have recently been developed. Thus, while the fundamental mechanisms regarding skin aging are known, new substances should be considered for introduction into dermatological treatments. Herein, we describe melatonin and its metabolites as potential “aging neutralizers”. Melatonin, an evolutionarily ancient derivative of serotonin with hormonal properties, is the main neuroendocrine secretory product of the pineal gland. It regulates circadian rhythmicity and also exerts anti-oxidative, anti-inflammatory, immunomodulatory, and anti-tumor capacities. The intention of this review is to summarize changes within skin aging, research advances on the molecular mechanisms leading to these changes, and the impact of the melatoninergic anti-oxidative system controlled by melatonin and its metabolites, targeting the prevention or reversal of skin aging.

## 1. Introduction

The skin is the most complex and multifunctional self-regulating organ. Facing the environment, the cutaneous barrier protects the body from external stressors and is essential for cutaneous and overall body homeostasis [1,2,3,4]. Additionally, the skin, together with the hypodermis (subcutaneous fat), is both a source and a target for many hormones and neuromediators [5,6,7,8,9,10,11,12,13,14,15], making it an independent and fully functioning peripheral endocrine organ [5,16]. Important mechanisms of the skin in maintaining homeostasis and protecting the whole body include the regulation of oxidative stress mechanisms and circadian rhythm [17]. The skin has its own peripheral circadian machinery, working either along with the central circadian clock or autonomously [18]. Like other organs, the skin also follows a rhythmicity in the production of bioactive molecules and sebum, and a periodicity in hydration, surface pH, skin temperature, capillary blood flow, etc. [19,20,21]. To counteract oxidative stress, the skin produces several protective molecules, including melatonin, vitamin D, and melanin [22,23,24,25,26,27]. Unfortunately, the endogenous antioxidant capacity of the skin is reduced with age and oxidative damage accumulation during aging, making the aged skin more vulnerable to environmental insults, especially ultraviolet (UV) radiation, air pollutants, and pathogens.

Biological aging is a natural phenomenon accompanied by a progressive loss of functional capacity, physiological integrity, and morphological features of the organism. The chronobiological functioning of the skin influences its aging. Mechanisms underlying the aging process include oxidative stress, mitochondrial dysfunction, disruption of circadian rhythms, inflammation, proteostasis, telomere attrition, genomic instability, epigenetic alterations, and a decreased capacity for tissue repair [28,29]. Circadian clocks are vital to human health through the rhythmic activity of physiological and neuro-endocrine functions. Aging is associated with a decline in circadian rhythm and a dampening of circadian gene expression [30] that can augment oxidative stress through an increased generation and accumulation of reactive oxygen species (ROS) [31]. Melatonin as well as vitamin D can regulate the cutaneous redox state and circadian rhythm [17,32].

The indolic hormone melatonin, which is released by the pineal gland, orchestrates circadian rhythms and sleep promotion [33,34]. There are also extrapineal tissues, like human skin, where it is synthesized [22,35] and works on site as a multifunctional molecule. Cutaneous melatonin production also follows rhythmicity, with the highest levels of cutaneous melatonin in the evening [36]. Melatonin produced in the skin exerts a protective effect against cutaneous damage caused by external factors [37]. Melatonin and its metabolites, including indolic derivatives such as 6-hydroxymelatonin and 2-hydroxymelatonin and kynuric metabolites such as AFMK and AMK, can limit oxidative stress via the scavenging of toxic radicals and inhibition of their generation, especially at the mitochondrial level [22,23,35,38,39,40,41,42]. Additionally, melatonin demonstrates potent antioxidant properties through its capacity to stimulate the production of antioxidant enzymes [43]. Furthermore, melatonin can also ameliorate DNA damage caused by environmental factors [40] and has anti-inflammatory [44] and anti-apoptotic effects [45,46]. This pleotropic regulatory action of melatonin and its metabolites on the skin makes them powerful anti-aging molecules. Because the synthesis of peripheral melatonin decreases with aging, the endogenous cutaneous melatonin production could be amplified with topical application of melatonin, which is considered as an effective photoprotective agent [37,47] and a very promising anti-aging strategy [48].

## 2. Skin Aging

### 2.1. Natural Process of Skin Aging

The aging of the skin is a natural and genetically determined process with progressive morphological and functional alterations, which are influenced by the total exposure to both the environmental and internal factors over the human lifespan [49]. The physiological maturing process results in most of the phenotypic changes of aging observed in all skin areas, including the appearance of fine wrinkles, atrophy with reduced elasticity, and prominent dryness often accompanied by pruritus. However, they vary among different anatomical regions and within different ethnicities [50,51].

Chronological (physiological) skin aging is caused mainly by an imbalanced endocrine circadian rhythmicity, with a hormonal decline and changes in gene expression with advancing age [51,52,53,54]. Aging affects proopiomelanocortin (POMC) and POMC-derived peptides, especially melanocortin receptor 1 (MC1R) and MC2R agonists, implicating their role in the general process of skin aging [55]. The single nucleotide polymorphisms (SNPs) of the *MC1R* gene are significantly linked to perceived facial age [56]. Putative functionally relevant SNPs can also affect the other pigmentation-related genes (e.g., *IRF4*, *ASIP*, *BNC2*) [57]. These genetic variations identified in skin color genes contribute to facial pigmented spots during aging through pathways independent of the melanin production [58]. Recently, the association of variants in *IRF4*, *MC1R,* and *SLC45A2* with skin wrinkling was confirmed in more ethnic groups [59]. The same study, using Latin Americans of a mixed continental ancestry cohort, reported genetic variations in two new candidate genes, *VAV3* and *SLC30A1*, associated with facial skin wrinkling and mole count, respectively [59]. Epigenetic mechanisms are also implicated in the direct regulation of the homeostasis and regeneration of the aged skin [60].

The process of aging involves the excessive senescence of keratinocytes, fibroblasts, and melanocytes over time, with accumulation contributing to decreased cutaneous regenerative potential and skin aging (Figure 1) [61,62,63,64]. Senescent skin cells are metabolically active and secrete diverse pro-inflammatory cytokines, chemokines, proteases, and growth factors in a state known as the senescence-associated secretory phenotype (SASP) [65]. This SASP’s state plays a role in the functional decline of physiologically aged skin [66,67]. With accelerating age, the immune system also undergoes senescence that can cause dysregulation of immune responses and possible impairment of the cutaneous immunological defense and adaptive capacity [68,69,70]. Indeed, the main cellular perturbations in the skin inducing senescence are inflammation and oxidative stress.

In chronological aging, ROS are produced through cellular oxidative metabolism, where mitochondrial dysfunction has an impact. Accumulating evidence supports a strong link between a decline in mitochondrial quality and function and the aging process [71,72]. Mitochondria also undergo aging, characterized by significant increase in ROS generation, a decrease in oxidative capacity and antioxidant defense, and a reduction in oxidative phosphorylation and adenosine triphosphate (ATP) production. This age-related impaired function of mitochondria further enhances mitochondria-mediated apoptosis, which contributes to an increase in the percentage of apoptotic cells [73]. An important target of ROS is mtDNA, in which damage and decline in function result in further enhancement of ROS production [74,75].

### 2.2. Environmentally Induced Skin Aging

Physiological aging is influenced by environmental stressors which can drive the premature aging of the skin. The most prominent external factors are ultraviolet radiation (UV) [76,77,78] and ambient pollutants [79,80,81,82]. Long-term exposure of the skin to these environmental insults stimulates ROS and reactive nitrogen species (RNS) production, and generates oxidative stress [83,84]. Further, they contribute to premature cutaneous aging, demonstrated by deep-wrinkle formation, sagging, and pigmentation affecting mainly exposed areas like the skin on the face, neck, head, and hands [85,86]. Chronic exposure can also cause an impairment of the epidermal barrier function [87] and alterations in the skin microbiome [88], leading to significant morbidity [70,89].

UVR is the most widely recognized harmful environmental factor that affects cutaneous biology and contributes to photodamage. The superposition of the solar damage on the physiological aging process leads to chronic inflammation, impaired regenerative capacity, and photoaging, which correlates with enhanced cancer risk [76,90,91,92]. Both ultraviolet (UV)A (315–400 nm) and UVB (280–315 nm) wavelengths have been shown to contribute to photoaging, either by imbalanced ROS/RNS production or by direct DNA damage [84,91]. Indeed, UVA is considered to play a major role in the skin´s aging process. UVA constitutes more than 80% of total daily UV irradiation and penetrates 5–10 times deeper into the reticular dermis, significantly damaging the extracellular matrix (ECM) in comparison with UVB [91]. This UVA effect is based on an increase of the transcription of matrix metalloproteinases (MMPs), especially the collagenolytic enzyme MMP-1 in dermal fibroblasts, causing massive collagen degradation and procollagen inhibition. Loss of balance between the essential tissue-specific inhibitor (TIMP1) of MMPs and MMP-1 can contribute to wrinkle development [93]. Thus, MMP-1 serves as an important regulator in photoaging [94]. Additionally, UVA exposure stimulates the activity of elastase and hyaluronidase and inhibits hyaluronan synthesis, thereby altering the composition of proteoglycans and glycosaminoglycans in the dermis [84,95]. Chronic UVR (mainly UVA exposure) is also indirectly related to photoaging and photocancer due to an excessive generation of ROS and RNS, which can disrupt both the nuclear and mitochondrial DNA [96,97].

UVB can penetrate only through the epidermis but is biologically more active [76,98]. UVB radiation absorbed by DNA and RNA directly induces a formation of cyclobutane pyrimidine dimers (CPDs) and other photoproducts in keratinocytes [99]. Further, DNA photolesions may trigger various typical solar signature mutations in specific genes, including the tumor suppressor gene p53 [100,101]. The induced UVR accumulation of p53 protein in the nucleus in turn activates the transcription of genes responsible for cell cycle arrest, allowing DNA repair. P53 accumulation also results in an induction of apoptosis of the cells with unrepaired DNA damage [102].

The exposure of the skin to environmental air pollutants and their negative impact is of growing concern [103]. Their prolonged exposure can alter the skin homeostasis and has been associated with skin aging and other cutaneous pathologies [49,79,81]. Additionally, air pollutants, persistent organic pollutants, and heavy metals can behave like endocrine-disrupting chemicals (EDCs) [104]. Ozone from the smog and particulate matter (PM) in contact with the skin is capable of stimulating the production of ROS and generates oxidative stress, leading to typical phenotypic features of premature aging, including pigment spots and deep nasolabial folds [105,106]. Moreover, ultrafine particles (<0.1 μm) can penetrate tissues and localize in the mitochondria, resulting in mitochondrial damage from the oxidative processes [107]. Moreover, the chronic photopollution stress of the skin may aggravate UVR-mediated skin aging [108].

Generally, environmentally induced premature skin aging is mainly driven by oxidative events. The mitochondria can generate about 90% of the intracellular ROS and are thus considered a main source of free radical production [109,110]. In addition to mitochondrial ROS, another important source of free radicals is the nicotinamide adeninedinucleotide phosphate (NADPH) oxidase system, which also plays a key role in triggering of oxidative stress. Because of the oxidative stress, increased levels of highly reactive free radicals promote lipid peroxidation, protein oxidation, genomic and mitochondrial DNA (mtDNA) damage, and depleted enzymatic and non-enzymatic antioxidant defense systems of the skin [111,112,113,114]. The accumulation of ROS/RNS dysregulates cell signaling pathways, alters cytokine release, and leads to inflammation. Indeed, the overproduction of ROS activates mitogen-activated protein kinases (MAPKs) and transcription factors such as nuclear factor-κB (NF-κB), nuclear factor erythroid 2-like (Nrf2), and c-Jun-*N*-terminal kinase (JNK) [115,116,117]. Levels of redox-sensitive activator protein-1 (AP-1) and NF-κB are found to be elevated within hours after exposure to low-dose UVB. Both NF-κB and AP-1 contribute to wrinkle formation and inflammation, and play crucial roles in accelerated skin aging. The up-regulation of AP-1 suppresses the transforming growth factor β (TGF-β) receptors, which further blocks procollagen synthesis [118]. Furthermore, activated AP-1 stimulates collagen breakdown by MMPs and triggers the main activator of the inflammatory response, NF-κB. NF-κB pathways are involved in a regulation of tissue homeostasis and aging [119,120]. ROS-triggered activation of NF-κB drives an elevation of proinflammatory cytokines (IL-1, IL-6, and TNF-α) and MMPs, and decreases TGF-β and collagen type I synthesis [119]. Additionally, enhanced NF-κB expression was found in mitochondrial DNA (mtDNA)-depleter mice, confirming that NF-κB signaling is a decisive mechanism contributing to the skin and hair follicle pathologies [114]. Solar-induced inflammation is also associated with deficiency of the aging suppressor hormone klotho [121]. Klotho is a transmembrane protein, and its function is possibly mediated through the toll-like receptor 4 (TLR4)/NF-κB axis signaling pathway [122]. Additionally, klotho can prevent NF-κB translocation, leading to an inhibition of the pro-inflammatory NF-κB pathway.

The endogenous Nrf2 is essential for cutaneous protection from oxidative insults and for regulating the redox balance during skin aging [116,123]. UVA, due to its longer wavelength, reaches the dermal fibroblasts in vivo, where it stimulates Nrf2-mediated antioxidant gene expression. Unlike UVA, UVB does not activate Nrf2 in skin cells or even appear to have an inhibitory effect [124,125]. However, vitamin D_3_ derivatives, products of UVB action, can activate Nrf2 signaling [125]. Thus, Nrf2 and its downstream signaling play a crucial role in photoprotection [117,126].

Recently, some Sirtuins (SIRTs) have gained attention due to their epigenetic ability to deacetylate histone and nonhistone targets, modulating the expression of genes implicated in the oxidative stress response and apoptosis [127]. The expression of SIRT1 and SIRT6 is found to be significantly reduced in aged human fibroblasts [128]. In addition, UVB irradiation reduces the expression of SIRT1 [129]. Furthermore, down-regulation of SIRT1 leads to an increase in MMPs and NF-κB activity. Thus, the activation of SIRT1 proves to have beneficial impact on both chronological and premature skin aging [127].

## 3. Melatonin and Aging

### 3.1. An Overview of the Synthesis, Metabolism, and Function of Melatonin

The phylogenetically ancient molecule melatonin (*N*-acetyl-5-methoxytryptamine) is widely distributed in nature [130,131,132] and can be formed almost in all living organisms, including plants [133,134,135,136]. Melatonin was first isolated and identified in bovine pineal gland by the dermatologist Aaron Lerner at al. in 1958 [137]. Lerner, together with his colleagues, was also the first to identify melatonin’s chemical structure and its action as a lightening agent in melanophores counteracting the α-melanocyte stimulating hormone (α-MSH) [138]. Historically, in mammals, this indolamine was thought to be uniquely released by the pineal gland, playing a major role in the regulation of circadian day–night rhythms and seasonal biorhythms [33,139]. Pineal-released melatonin can be measured at lower concentrations in the blood than in the cerebrospinal fluid (CSF) of the third ventricle of the brain, suggesting its role as a protector of the brain against oxidative stress [140,141]. Later, extrapineal sites of melatonin production were established. Thus, melatonin is also synthesized in numerous peripheral tissues such as the bone marrow, retina, lens, cochlea, lungs, liver, kidney, pancreas, thyroid gland, female reproductive organs, and finally the skin [14,15,22,142,143,144,145,146]. Indeed, the synthesis of melatonin is a multistep process that first starts with hydroxylation of L-tryptophan to 5-hydroxy-tryptophan (5(OH)tryptophan, catalyzed by tryptophan hydroxylase [147,148,149]. Further 5(OH)tryptophan is decarboxylated to serotonin, which is subsequently transformed to *N*-acetylserotonin (NAS) by the enzyme arylalkylamine *N*-acetyltransferase (AANAT) [150,151]. Furthermore, it has been found that serotonin can be acetylated to NAS by alternative enzymes including arylamine *N*-acetyltransferase [152,153,154,155,156]. The last step in the synthesis is a conversion of NAS to melatonin by hydroxyindole-*O*-methyl transferase (HIOMT) [157].

The levels of melatonin are regulated by its rapid metabolism in the liver or directly at the site of its synthesis in peripheral organs [158]. In the classical hepatic metabolism, CYP450 enzymes (CYP1A1, CYP1A2, and CYP1B1) degrade circulating melatonin to 6-OH-melatonin [159,160]. Melatonin can also be demethylated in the liver to NAS by CYP2C19 or CYP1A, which represents a minor microsomal pathway [161,162]. Through the alternative indolic pathway, melatonin is deacetylated by liver aryl acylamidase to 5-OH-tryptamine, which is further deaminated by monoamine oxidase A [163]. The metabolism of melatonin through kynuric pathway begins with the formation of *N*^1^-acetyl-*N*^2^-formyl-5-methoxykynuramine (AFMK) in a peroxidase-like reaction. Further AFMK is deformylated to *N*^1^-acetyl-5-methoxykynuramine (AMK) [164,165]. In mitochondria, an additional route of melatonin metabolism to AFMK by cytochrome *C* oxidation has also been described [166]. In the skin or skin cells, melatonin is metabolized rapidly through its 6-hydroxylation, through the indolic and kynuric pathway, and through non-enzymatic processes including phototransformation induced by UVB, UVA, and reactive oxygen species [167,168,169]. The main products of melatonin metabolism in the epidermis are 6-hydroxymelatonin, AFMK, AMK, 5-methoxytryptamine, 5-methoxytryptophol, and 2-hydroxymelatonin. These products accumulate in the epidermis at detectable concentrations [170,171].

The widespread melatonin distribution during evolution has made it as a vital multifunctional hormone, with remarkable essential functions [34,172]. The complex action of melatonin includes its work as a regulator of the circadian clock, a neurotransmiter and hormone, a metabolic modulator, and a modifier of cell response and cytokine release [173,174,175,176,177]. It also regulates the functions of many peripheral organs [174,178] and exerts oncostatic [179,180,181,182,183,184] and anti-aging capacity [48,185]. Many regulatory effects of melatonin on cardiovascular, endocrine, reproductive, and immune systems are mediated via specific melatonin 1 (MT1) and MT2 membrane receptors [19,186]. Melatonin, by interacting with MT1 and MT2, has been found to limit weight gain [176,187,188]. Melatonin can inhibit adipogenic differentiation and together with vitamin D, exhibits a negative regulation of adipogenesis in adipose-derived stem cells (ADSCs). It was recently found that melatonin significantly inhibited the transcription of specific adipogenesis-orchestrating genes, such as *aP2* and peroxisome proliferator-activated receptor γ (*PPAR-γ*), as well as adipocyte-specific genes including lipoprotein lipase (*LPL*) and acyl-CoA thioesterase 2 (*ACOT2*). Moreover, melatonin and vitamin D can modulate ADSCs through the upregulation of epigenetic regulatory genes like histone deacetylase 1 (HDAC1), SIRT1, and SIRT2 [189].

Melatonin can also inhibit the effects of estrogens [190], and exhibits cardioprotective [191,192] and anticonvulsant activity [193]. MT1 and MT2 are also important for protection of the skin against environmental stressors, aging, and cancerogenesis [179,194]. Moreover, often the melatonin level inversely correlates with an increased risk of cancer development. Of note, the blockage of melatonin receptors can impair the p53-dependent DNA damage response [195]. The antioxidant ability of melatonin relays the indirect receptor-mediated action, likely by the stimulation of antioxidant enzymes, SIRT3, and others [43,196]. Melatonin works also through non-receptor mediated mechanisms such as the direct scavenging of variety of reactive species (both ROS and RNS) to counteract oxidative stress [39,41,130,197,198,199]. In addition to its high antioxidant potential, receptor-independently, melatonin serves as a mitochondrial protector [200] and anti-inflammatory agent [201]. Some of the protective properties of melatonin are shared with its kynuric metabolites AFMK and AMK [178,202,203].

### 3.2. Protective Role of Melatonin in Systemic Aging 

The “free radical theory of aging” has been discussed for over 50 years [204,205,206]. At the subcellular level, mitochondria are the major source for generation of a highly reactive and destructive species like peroxynitrite and the hydroxyl radical [207]. Their excessive production, resulting in enhanced mitochondrial oxidative stress and mtDNA mutations, occurs along with human aging and age-related pathologies [208,209,210]. Some intracellular enzymes outside the mitochondria (e.g., xanthine oxidase, monoamine oxidase, NADPH oxidases) also impact on ROS production with advancing age [211,212,213]. Disturbances in mitochondrial redox balance promote cellular senescence and thus the mitochondria impairment determines the rate of aging [214]. Recently, it has been thought that most mtDNA mutations are caused by replication errors of mtDNA polymerase [215]. During aging, such defects in mtDNA replication machinery together with a failure of their repair might cause an accumulation of mutations with further mitochondrial dysfunction and augmentation of oxidative damage.

Since free radicals abundantly are generated in mitochondria in aging, molecules that reduce their mitochondrial production or detoxify them may slow the rate of systemic aging. Melatonin is such a molecule, and its role in aging has been on the focus of many scientists in the last 20 years [42,216,217,218]. It was found that surgical pinealectomy of young rats resulted after time in accelerated oxidative damage in multiple tissues due to circadian disruption, and melatonin-deficient animals aged more rapidly [219].

While dysfunctional mitochondria contribute to the aging process [220], melatonin can maintain optimal mitochondrial physiology [42,221,222]. Melatonin concentrations are found at higher levels in mitochondria than in other cellular organelles, suggesting its significant role as a mitochondrial-targeted molecule involved in mitochondrial processes [42,200]. The multiple beneficial protective actions of this indolic hormone at the mitochondrial level are well documented [223]. Melatonin can limit age-related oxidative stress directly by scavenging ROS/RNS [41,224] and by indirect activation of mitochondria-located superoxide dismutase (SOD2) [225]. Through the stimulation of mitochondria’s localized SIRT3, melatonin prompts the deacetylation and activation of SOD2. The activation of antioxidant enzymes involved in the SIRT3/SOD2 signaling pathway by melatonin reduces mitochondrial oxidative damage and cytochrome *C* release, thus reducing mitochondria-related apoptosis [196,226]. Indeed, melatonin maintains the optimal mitochondrial membrane potential and preserves mitochondrial function not only by quenching free radicals [198] but also by inhibiting the mitochondrial permeability transition pore (MPTP) [227], activating uncoupling proteins (UCPs), and regulating mitochondrial biogenesis and dynamics [228].

Generally, melatonin may act as both pro- and anti-inflammatory molecule in a context-dependent fashion [201,229,230]. In aging, melatonin preferentially exerts anti-inflammatory actions on aging-related low-grade inflammation. Melatonin stimulates SIRT1, and their anti-inflammatory activities overlap during the process of aging [231]. SIRT1, functioning as an epigenetic aging regulator, alleviates the inflammation by downregulating TLR4, which mediates pro-oxidant effects through the NF-κB signaling pathway [229]. Melatonin, by an inhibition of either TLR-4 and the toll receptor-associated activator of interferon (TRIF), can suppress the release of several pro-inflammatory cytokines like TNFα, IL-1β, IL-6, and IL-8 [232,233].

To summarize, melatonin, with its capacity to mitigate oxidative stress, protect mitochondrial functions, modulate the immune system, reduce inflammation, enhance circadian rhythm amplitudes, and exhibit neuroprotection, beneficially results in retarding the process of aging [174,216,234,235,236,237,238,239,240].

## 4. Melatonin, Its Metabolites and Skin Aging

### 4.1. Overview of Cutaneous Melatoninergic System

Melatonin is synthesized and metabolized in the skin. The ability of the mammalian skin to synthesize melatonin from serotonin through NAS was first published in 1996 [241]. Follow-up studies have provided the evidence that human skin, as well as normal keratinocytes, melanocytes, and melanoma cells, can endogenously produce melatonin [13,14,15,22,242]. Moreover, the skin cells express the essential enzymes for transforming tryptophan to serotonin and eventually to melatonin, like tryptophan hydroxylase (TPH1—all skin cells; TPH2—melanocytes and dermal fibroblasts) [13,14,23,243], AANAT/serotonin *N*-acetyltransferase (SNAT) and NAT [154,155], and HIOMT/*N*-acetylserotonin-methyltransferase (NASM) [13,14]. Cutaneous serotonin can be acetylated to NAS by both AANAT and NAT [13,152,156]. Hair follicles also generate melatonin and express its functional receptors [244]. Recently, the concentrations of melatonin and its metabolites in the human epidermis were quantified by liquid chromatography–mass spectrometry (LC-MS) [170,171]. The level of epidermal melatonin varies depending on race, gender, and age. Kim et al. measured the highest concentrations of melatonin among African Americans and elderly Caucasians. The levels of its kynuric metabolite AFMK were significantly higher in Caucasian males, whereas AMK demonstrated higher concentration in African Americans than in Caucasians [171]. The accumulation of AMK in the epidermis suggests the cutaneous transformation of AFMK to AMK.

Melatonin in the skin undergoes rapid metabolism in vivo through either the indolic and kynuric pathways, with 6-hydroxymelatonin being a major metabolite [168,169]. Indeed, all metabolites of melatonin, including the final kynuric metabolites AFMK and AMK, are present in the epidermal cells and can potentially affect their mitochondrial functions [35,245]. Exposure of human skin to UVB can induce melatonin metabolism, leading to the generation of antioxidant metabolites AFMK and AMK in human keratinocytes [167,169]. The photo-induced melatonin metabolites further form a very potent anti-oxidative cascade. This cascade has been defined as the melatoninergic anti-oxidative system (MAS) of the skin [13,167].

Melatonin and its metabolites are essential for the regulation of many skin functions, including cutaneous pigmentary [13,246], adnexal [244,247,248], barrier [23,40,168], and immune [173] functions. They also protect the skin against external and internal insults (Figure 2), and possess an oncostatic potential in melanoma cells [180,249]. Unlike melatonin, AMK does not inhibit tyrosinase activity and has no significant effect on melanogenesis [170]. Some but not all the phenotypic effects of melatonin are mediated via interaction with membrane bound G-protein-couple MT1 and MT2 receptors. MT1 has widespread localization, mainly in the epidermis (stratum granulosum, stratum spinosum, upper and inner root sheath of hair follicles) [19,22], whereas MT2 is often found in hair follicles and blood vessels, with lower expression or absence in the epidermal cells [13,244]. The expression of MT2 in hair follicles makes them a possible target for hair growth regulation by melatonin [248]. “MT3 receptors” have been also detected in keratinocytes, melanocytes, and fibroblasts; however, their role requires clarification [179]. Nuclear retinoic orphan receptor α (RORα) has been found to be expressed in skin cells but it is not a receptor for melatonin, being identified as a receptor for sterols and secosteroids [250,251]. Melatonin regulation of mitochondrial functions is predominantly receptor-independent and requires high concentrations which can be achieved by an efficient on-site production and/or topical melatonin application.

### 4.2. Role of Melatonin and Its Metabolites in Attenuation of Photoaging

Although skin has a well-equipped powerful antioxidant system to counteract oxidative stress, chronic exposure to UVR with its excessive ROS production can overcome the endogenous antioxidant defense of the skin, resulting in damage and premature aging in a process known as photoaging. Melatonin is one of the protective molecules biosynthesized at high concentrations in mitochondria of the skin cells to incapacitate ROS by electron donation and RNS by nitrosylation reactions [199,252,253]. Melatonin can prevent the formation of highly reactive free radicals by reducing the superoxide anion radical (O_2_•¯) in a process referred to as radical avoidance [228,254]. The positional advantage of melatonin increases its ability to immediately scavenge the toxic free radicals formed in abundance in mitochondria, mainly by UVA but also by UVB irradiation [198,245]. Melatonin may additionally stimulate enzymes that are able to degrade the weakly reactive ROS [130,255]. It is important to note that the most harmful species (hydroxyl radicals and peroxynitrite) are not degraded by enzymes. They can only be removed by a direct highly efficient scavenger like melatonin [256,257,258]. The reaction of melatonin with hydroxyl radical initiates the formation of 2-OH-melatonin and 4-OH-melatonin, which are further metabolized to AFMK and by arylamine formamidase or catalase to AMK [196,202]. The effective toxic radical scavenging mediates the reduction of ROS-generated oxidative stress.

In normal and diabetic human dermal fibroblasts, melatonin can stimulate SOD, catalase (CAT), and glutathione peroxidase (GPx), and promote glutathione (GSH) production [259]. Indeed, through activation of MT1/MT2, melatonin up-regulates the expression of antioxidant genes in irradiated cells [43,245,260].

The molecular mechanism of the indirect antioxidant action of melatonin with regard to the activation of phase-2 antioxidant enzymes has recently been established in UV-exposed human keratinocytes [254] and UVB-treated melanocytes [194]. It was found that melatonin stimulated NRF2 expression and induced its translocation to the nucleus, leading to enhanced gene expression of its target enzymes including γ-glutamylcysteine synthetase (γ-GCS), heme oxygenase-1 (HO-1), and NADPH:quinone dehydrogenase-1 (NQO1) [254]. The up-regulation by the melatonin/NRF2-dependent pathway supports the elevated antioxidant response of both keratinocytes and melanocytes against UVB-induced oxidative stress. [37,47,194]. Moreover, Nrf2 activation protects scalp hair growth against oxidative damage [261]. The ability of melatonin to attenuate UVA/UVB-induced alterations and to prevent the further photodamage has also been demonstrated in fibroblasts (Figure 3) [262,263]. In addition, it was found that melatonin can reduce the number of 8-hydroxy-2′-deoxyguanosine (8-OHdG)-positive cells, a marker of oxidative DNA damage [23,260]. Thus, by being a broad-spectrum antioxidant and amphilic molecule, melatonin can penetrate membranes and can also attenuate UVR-induced lipid peroxidation, protein oxidation, and mitochondrial and DNA oxidative damage [23,35,37,41,47,264]. The other protective capability of melatonin is to counteract UVR-induced alterations in the mitochondrial ATP synthesis, plasma membrane potential, and pH in human keratinocytes [46,254,265].

Importantly, melatonin possesses an advantage when compared with other antioxidants, since melatonin exerts not only a potent antioxidant capacity but most of its metabolites are antioxidants as well [168,202]. Whereas classical antioxidants (vitamins C and E) scavenge a single radical, melatonin’s antioxidant cascade detoxifies many toxic radicals. Moreover, accumulating evidence supports the reciprocal interaction between melatonin and NAS in mitochondria that would amplify the detoxification process [169,178,245]. In addition, melatonin activates cytochrome *C* in mitochondria [159], which possibly mediates the formation of final kynuric metabolites, which are even better free radical scavengers than melatonin itself [202,203,266]. AFMK and AMK generated non-enzymatically can accumulate in the skin [243]. However, AMK can disappear very quickly through oxidation and interactions with RNS [169].

Melatonin and its derivatives (6-hydroxymelatonin, NAS, AFMK, AMK, and 5-methoxytryptamine) have the capacity to protect keratinocytes and melanocytes against UVB-induced cell damage [23,37,194]. They not only reduce the formation of CPDs and 6–4 pyrimidine–pyrimidone photoproducts, but also induce the repair of DNA damaged by UVB. It has been demonstrated that the topical application of melatonin and AFMK can prevent DNA damage and apoptosis in human and porcine skin ex vivo [47]. Furthermore, the pre-incubation of full-thickness skin and normal human keratinocytes with melatonin suppressed the UVB-mediated inflammatory and apoptotic effect, as measured by heat shock protein 70 expression, expression of pro-inflammatory cytokines (IL-1*β*, IL-6), and the pro-apoptotic protein caspase-3 [267]. The photoprotective potential of topically administrated melatonin has been shown in many clinical studies. Thus, treatment of the skin with exogenous melatonin before and after sun exposure attenuates UVR-induced erythema and oxidative stress [268]. The effect is greater when the cutaneous application of melatonin cream occurs prior to UVB exposure [269]. Sunscreens supplemented with melatonin could be used to prevent a skin photoaging and photocancerogenesis [270].

One potential anti-wrinkle mechanism of melatonin was studied by Sung-Hoon Kim’s group [44]. They found that melatonin, by reducing ROS production, diminished MMP-1 expression and increased collagen XVII expression in HaCaT keratinocytes exposed to UVB. Furthermore, in the same study melatonin was shown to reduce the transepidermal water loss (TEWL) on the skin of hairless mice 8 weeks after UVB irradiation [44]. A clinical study also demonstrated a significant reduction of facial redness and wrinkles, and an improvement in the epidermal barrier function by using a night serum combination of melatonin, vitamin C (lipophilic and non-oxidizable form), and a polyphenol compound (bakuchiol) with retinol-like properties [271]. Additionally, the same night serum containing melatonin has been shown in vitro to increase filaggrin levels in keratinocytes, and collagen I and III in dermal fibroblasts, as well as to reduce the formation of apoptotic sunburn cells in UV-exposed skin ex vivo. [272].

The above findings confirm the clinical potential of melatonin as a broad-range photoprotector which can have a great impact on an attenuation of the premature skin aging and the improvement of the hallmarks of photoaged skin [14,47,273,274].

### 4.3. Role of Melatonin and Its Metabolites in the Attenuation of Pollution-Induced Skin Aging

Environmental air pollutants promote mitochondrial dysfunction and oxidative damage due to excessive ROS generation, potentially resulting in prematurely aged skin and skin cancer [107,108]. Melatonin can restore the mitochondrial function and maintain the mitochondrial homeostasis [275]. It can reach the mitochondria by crossing the cell membranes, and it can also be synthesized in the mitochondrion. High concentrations of melatonin in mitochondria (endogenously produced or exogenously applied) can reduce oxidative damage, preserve mitochondrial respiration, limit mitochondria-related apoptosis, increase mitochondrial membrane potential and ATP production, and regulate mitochondrial biogenesis and mitophagy (removal of the damaged mitochondria).

It has been proposed that SIRT1, which can be stimulated by melatonin as well, plays a crucial role against pollutant-related premature skin aging. The up-regulation of SIRT1 could downregulate the MMP-1 and MMP-3 involved in the collagen breakdown, and it could decrease inflammation through inhibition of NF-κβ signaling [127].

The use of creams containing melatonin, carnosine, and Helichrysum italicum extract on skin explants exposed to a mixture of polycyclic aromatic hydrocarbons and heavy metals leads to a reduction in skin damage and irritation [276]. The study demonstrated a significant decrease in pollution-activated transcription factor aryl hydrocarbon receptors (AhR) and type I collagen in melatonin-treated explants.

Therefore, a skin care product containing melatonin would be a real “weapon” in the prevention of premature skin aging caused by urban pollutants, heavy metals, and cigarette smoke [277].

### 4.4. Possible Role of Melatonin in Modifying Natural Process of Skin Aging

The healthy aging of the skin is a complex multifactorial process that can be aggravated by an oxidative environment. With advancing age, the capacity of the skin to produce melatonin, the main direct- and indirect-acting antioxidant, diminishes, thus contributing to a decline in the endogenous protective MAS. The decreased levels of melatonin with age are accompanied with dysregulation of the circadian rhythm. Additionally, an age-dependent decrease in MT1 receptors is found in aged human fibroblasts [278]. The reduction in MT1 receptors along with a reduced melatonin level results in enhanced skin cellular damage and phenotypical signs of aging.

Therefore, the administration of exogenous melatonin would be a good anti-aging strategy. Orally supplemented melatonin appears in rather low levels in the blood due to prominent first-pass degradation in the liver, thus limiting skin access [14]. Topically applied melatonin may penetrate the stratum corneum and form a depot there due to its distinct lipophilic chemical structure [279]. The application of melatonin on the skin is a very good option for retarding the aging process and reducing the hallmarks of skin aging. The cutaneous application of melatonin is efficacious and safe way to improve the clinical signs of aging (wrinkles, TEWL and hydration, skin roughness, sagging, etc.) [186]. Clinically, it is better to apply melatonin at nighttime when the skin permeability is higher and because melatonin can mimic its endogenous production and effects.

With its pleotropic protective function of the skin, melatonin, with its proven beneficial anti-aging properties, could be considered as a therapeutic candidate for retarding skin aging and reversing cutaneous aging signs. Therefore, endogenous intracutaneous melatonin production, together with topically applied exogenous melatonin, is expected to provide the most potent defense system against cutaneous photodamage and multiple other pathological conditions that produce oxidative stress (e.g., in chronic skin inflammation, such as atopic dermatitis) [280]. Additionally, topical melatonin can be used for the treatment of androgenic alopecia in women [281].

## 5. Conclusions and Perspectives

Since the discovery of the strong antioxidant properties that melatonin possesses [137], a massive interest in terms of biological effects of melatonin in human and animal biology has evolved. It was shown that this indoleamine is an important bioregulator as well as a pluripotent and essential protective agent in many cells, tissues, and compartments of unicells, animals, and humans [22,216,282]. Melatonin exerts protective effects on cell physiology and tissue homeostasis, particularly in cutaneous cells exposed to UVR, which induces severe skin damage accompanied by oxidative stress or DNA damage. These intracellular disturbances are significantly counteracted or modulated by melatonin in the context of a complex intracutaneous melatoninergic anti-oxidative system with UVR-enhanced or UVR-independent melatonin metabolites. Therefore, endogenous intracutaneous melatonin production, together with topically applied exogenous melatonin or its metabolites, may be expected to represent a promising anti-oxidative defense systems against skin aging. Indeed, more research on appropriate in vitro, ex vivo, and in vivo models must be performed to substantiate the above idea. For example, we need to learn whether melatonin and its derivatives can affect the expression of senescence markers in the skin. It would be fascinating to explore the possibility as to whether cutaneous melatonin production is altered during skin aging. Moreover, it is crucial to know whether the expression of functional MTs in cutaneous cell types is impaired in aged skin, which could eventually limit the anti-aging effects of any topically applied type of melatonin. In summary, the key question is whether melatonin can be exploited therapeutically as a protective agent, as “a skin survival factor” with anti-genotoxic capacities, or as “the neutralizer” of pathological changes including skin aging and cancerogenesis. The efficacy of topically applied melatonin and its derivatives needs further evaluation in future clinical trials. Another important point that needs further investigation is the use of nanotechnologies and nanomaterials for the topical delivery of melatonin and its metabolites for skin rejuvenation or to preserve the young skin phenotype.

## Figures and Tables

**Figure 1 ijms-23-01238-f001:**
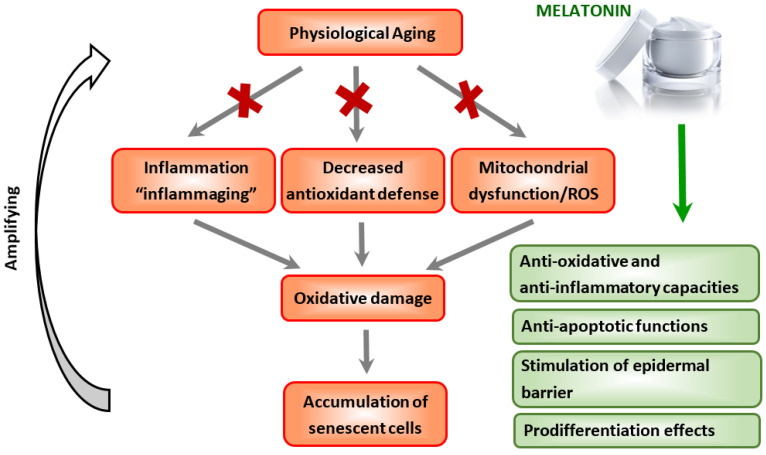
A possible role of melatonin in the prevention and treatment of physiological skin aging. Red crosses (✖) indicate protective action of melatonin against inflammaging, oxidative stress, and mitochondrial damage.

**Figure 2 ijms-23-01238-f002:**
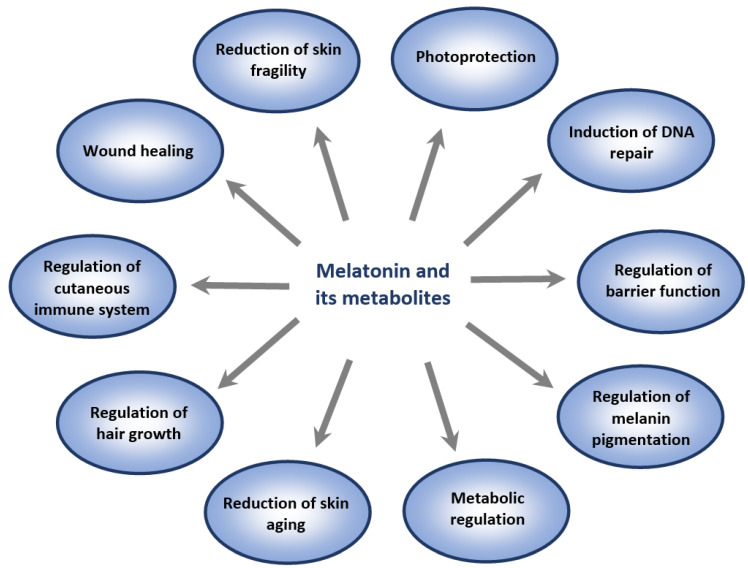
Overview of the pleiotropic effects of melatonin and its metabolites as major skin protectants. Melatonin directly or indirectly (via indolic and kynuric metabolites [35,169,245]) reduces deleterious intracellular changes including apoptotic disturbances or oxidative stress, while it maintains mitochondrial homeostasis.

**Figure 3 ijms-23-01238-f003:**
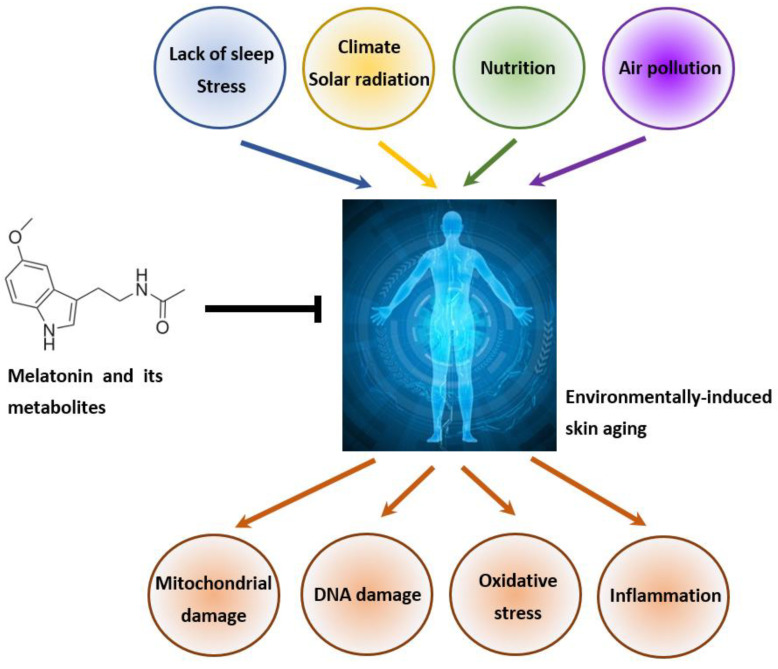
The protective role of melatonin and its metabolites against premature skin aging. Cutaneous melatonin can prevent mitochondrial and DNA damage, oxidative stress, and inflammation caused by environmental factors such as stress, solar radiation, poor nutrition, or air pollution.

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
