# Peer review of "Protective Role of Melatonin and Its Metabolites in Skin Aging"

_ijms, 2022, doi:10.3390/ijms23031238_

Round 1
Reviewer 1 Report
The manuscript “Protective role of melatonin and its metabolites in skin aging” is an interesting review where the authors discuss about the role of melatonin and its metabolites in counteracting skin aging thanks to its anti-oxidative, anti-inflammatory, immunomodulatory, and anti-tumour properties.
In the manuscript authors introduce the main changes occurring during skin aging and describe the mechanisms leading to aged skin. In particular authors analyse the role of melatonin related pathways in preserving the features of a young skin.
The review is well developed with innovative details. I suggest the publication of the manuscript after introducing the following minor changes:
- Lines 292-293: “Melatonin by interacting with MT1 and MT2 has been found to limit weight gain”. Other authors previously described that melatonin is able to interfere with stem cell differentiation toward the adipogenic phenotype. Stem cells have the important role to replace damaged and aged cellular elements, maintaining tissue homeostasis. The differentiation of stem cells towards the adipogenic phenotype could preclude their recruitment for the renewal of other skin cells, encouraging the appearance of an aging phenotype. Please discuss the role of melatonin in controlling weight gain, concerning the position of subcutaneous fat layer in the skin tissue and stem cell role.
- “Conclusions and Perspectives” section: authors hypothesize the possibility to use “topically applied melatonin and its derivatives” after further evaluation in clinical trials for topical applications. In the last years nanotechnologies and nanomaterials have been applied as novel perspectives for topical delivery in skin rejuvenation. Please discuss accordingly the possibility to apply this innovative tool to deliver melatonin for preserving young skin phenotype.
- The number of self citations is very high. Even though they can fit with the topic there are too many self-citations on the same subject. Please try to select the most important.
Author Response
Reviewer 1:
The manuscript “Protective role of melatonin and its metabolites in skin aging” is an interesting review where the authors discuss about the role of melatonin and its metabolites in counteracting skin aging thanks to its anti-oxidative, anti-inflammatory, immunomodulatory, and anti-tumor properties.
In the manuscript authors introduce the main changes occurring during skin aging and describe the mechanisms leading to aged skin. In particular authors analyse the role of melatonin related pathways in preserving the features of a young skin.
The review is well developed with innovative details. I suggest the publication of the manuscript after introducing the following minor changes:
Author’s reply: We thank the reviewer for positive comments, time and effort to improve our presentation.
- Lines 292-293: “Melatonin by interacting with MT1 and MT2 has been found to limit weight gain”. Other authors previously described that melatonin is able to interfere with stem cell differentiation toward the adipogenic phenotype. Stem cells have the important role to replace damaged and aged cellular elements, maintaining tissue homeostasis. The differentiation of stem cells towards the adipogenic phenotype could preclude their recruitment for the renewal of other skin cells, encouraging the appearance of an aging phenotype. Please discuss the role of melatonin in controlling weight gain, concerning the position of subcutaneous fat layer in the skin tissue and stem cell role.
Author’s reply: We thank the reviewer for the critique. We have addressed this issue on pages 6 and 7. The new sections are marked by tools.
- “Conclusions and Perspectives” section: authors hypothesize the possibility to use “topically applied melatonin and its derivatives” after further evaluation in clinical trials for topical applications. In the last years nanotechnologies and nanomaterials have been applied as novel perspectives for topical delivery in skin rejuvenation. Please discuss accordingly the possibility to apply this innovative tool to deliver melatonin for preserving young skin phenotype.
Author’s reply: We thank the reviewer for these comments. We have addressed this issue at the end of the section “Conclusions and Perspectives”.
- The number of self citations is very high. Even though they can fit with the topic there are too many self-citations on the same subject. Please try to select the most important.
Author’s reply: We have slightly reduced the number of self-citations, which has been replaced by new citation on adipogenesis. We believe that the reviewer agrees that the review should contain comprehensive reference list. Unfortunately, the authors of this review established many aspects of this field both experimentally and conceptually and it is difficult not to cite the original papers. We followed the rules of citations recommended by the previous editor of Endocrinology (Endocrinology, 151(1):1–3, 2010).
Reviewer 2 Report
In the work “Protective role of melatonin and its metabolites in skin aging”, the authors reviewed the mechanisms underlying skin aging, and how melatonin and its metabolites are involved as protecting agents. The review is very well written, very easy to follow, however, I’ve missed more figures explaining the role of melatonin and its metabolites. A few minor questions should be addressed:
- Page 2, lines 73 – 74 – Please elucidate which melatonin metabolites have the ability to limit oxidative stress. Authors could include melatonin and melatonin metabolites structures.
- Page 3, Figure 1 – Please improve Figure 1 to better explain the role of melatonin in the prevention and the treatment of physiological skin aging.
- Page 4 - “Environmentally induced skin aging” please add a Figure resuming all environmental effects on skin aging
- Page 8, lines 375-376 – please add a figure with the melatonin metabolism (both pathways) and how its metabolites are associated with skin functions and aging. This figure will increase the overall quality of the review and will give more information than Figure 2.
- Page 11, lines 498-499 – please check the sentence
Author Response
Reviewer 2:
In the work “Protective role of melatonin and its metabolites in skin aging”, the authors reviewed the mechanisms underlying skin aging, and how melatonin and its metabolites are involved as protecting agents. The review is very well written, very easy to follow, however, I’ve missed more figures explaining the role of melatonin and its metabolites. A few minor questions should be addressed:
Author’s reply: We thank the reviewer for these comments. We have addressed this issue at the end of the section “Conclusions and Perspectives”.
- Page 2, lines 73 – 74 – Please elucidate which melatonin metabolites have the ability to limit oxidative stress. Authors could include melatonin and melatonin metabolites structures.
Author’s reply: We thank the reviewer for these comments. It is now indicated on page 2 with referral to either indolic or kynuric configuration. We have also added citations as references to their structures.
- Page 3, Figure 1 – Please improve Figure 1 to better explain the role of melatonin in the prevention and the treatment of physiological skin aging.
Author’s reply: We thank the reviewer for these comments, and have revised figure 1 accordingly.
- Page 4 - “Environmentally induced skin aging” please add a Figure resuming all environmental effects on skin aging
Author’s reply: We would like to thank for this remark and we decided to enrich previously existed figure 3 with environmental factors. It is revised accordingly.
- Page 8, lines 375-376 – please add a figure with the melatonin metabolism (both pathways) and how its metabolites are associated with skin functions and aging. This figure will increase the overall quality of the review and will give more information than Figure 2.
Author’s reply: We appreciate the reviewer critique. However, we want to emphasize that we have published several schemes on melatonin metabolism in the skin and any new panel would represent only duplication what has been presented recently. The schemes are in the cited papers in the corresponding sections. We have corrected the legend to Figure 2 emphasizing chemical nature of metabolites with referral to proper citations.
- Page 11, lines 498-499 – please check the sentence
Author’s reply: We thank the reviewer for the critique. The sentence has been corrected.